# A Dynamic Adjustment Model of Cruising Taxicab Fleet Size Combined the Operating and Flied Survey Data

**Xiaofei Ye [1,\*], Min Li [1], Zhongzhen Yang [1], Xingchen Yan [2] and Jun Chen [3]**

[1]  Ningbo Collaborative Innovation Center for Port Trade Cooperation and Development, School of Maritime and Transportation, Ningbo University, Ningbo 315211, China; yangzhongzhen@nbu.edu.cn

[2]  College of Automobile and Traffic Engineering, Nanjing Forestry University, Nanjing 210037, China; xingchenyan.acad@gmail.com

[3]  School of Transportation, Southeast University, Nanjing 210096, China; chenjun@seu.edu.cn

\*  Correspondence: yexiaofei@nbu.edu.cn; Tel.: +86-15267859815.

**Abstract:** Due to the lack of adjustment index systems for taxi fleet sizes in China, this paper used the taxi operating datasets from Ningbo City and established a regression tree model to consider the endogenous indicators that affect taxi fleet sizes. Then, a dynamic adjustment mechanism of taxi fleet sizes was proposed by combining the exogenous and endogenous indicators. The importance of the exogenous and endogenous indicators was sorted using the Delphi method. The threshold value of each indicator was also given. The results indicated that (1) in the three-layer structure of the regression tree model, the mileage utilization had the strongest effect on the fleet size of taxis, and the F statistic was 63.73; followed by the average daily revenue of a single taxi, the average waiting time to catch a single taxi, the average operating time of a single taxi, and the revenue per 100 kilometers. The overall accuracy of the model was found to be valid. (2) When the mileage utilization was less than 0.6179 and the average daily revenue of a single taxi was less than 798.38 Yuan, the fleet size of cruising taxis was surplus and should be reduced by 362 vehicles. (3) When the mileage utilization was more than 0.6774 and the average waiting time to catch a single taxi was more than 259.09 seconds, the fleet size of cruising taxis was insufficient, and we suggest an increase of 463 taxis.

**Keywords:** Passenger management; Taxi; Fleet size; Regression tree; Threshold analysis

## 1. Introduction

As an important component of passenger transportation, the traditional taxicab provides trip services for people and occupies approximately 15–30% trips in the public transit systems [1]. With the rise of on-demand ride-hailing taxis like Didi and Uber, the traditional taxi market descended into chaos [2]. This has restricted the sustainable development of passenger transport in the cities of China. The conflicts between traditional taxi services and app-based ride-hailing services also increased the costs and caused serious social problems. For example, traditional taxi strikes against Didi and Uber happened in many cities in China. More than 5000 taxis were involved in Xi'an, but strikers put the number far higher. Some of the strikers harassed the ride-hailing drivers in certain cities. Normal taxi services were not available for passengers. Therefore, the Ministry of Transportation issued a regulation regarding the difference between traditional taxis and ride-hailing services and promoting the healthy development of the taxi market in China [3].

The regulation clearly announced that the local government should dynamically adjust the taxi fleet size (the number of the taxicabs). If the taxicab fleet size is not enough, where the passenger demand is more than the supply, it is difficult and inconvenient for the passengers to take a taxi for the trip. Otherwise, if the number of taxicabs is too high, negative effects, including traffic congestion, increases in passenger trip costs, and decreased taxi driver revenue, are generated.

How to regulate a reasonable taxi fleet size to satisfy the passenger demand without sacrificing the income of taxi drivers is an important premise of taxi fleet size adjustment. Meanwhile, both the operational efficiency and service level also need to be guaranteed. Unfortunately, accurate threshold values of indicators for adjusting the fleet size of taxicabs do not exist. It is necessary for the local government to establish a reasonable adjustment method for taxi fleet size and to alleviate the conflicts between the cruising taxis and Didi and Uber. A reasonable dynamic adjustment of the taxi capacity scale can increase the mileage utilization, alleviate the traffic congestion, and reduce the carbon emissions of taxicabs. Sustainable taxi industry is an important guarantee for promoting the green development of urban traffic. Additionally, as intra-city express service develop rapidly in China, taxicabs also tried to provide express, document delivery, baggage service and other freight services in city distribution. In order to gain more money, the taxicabs, such as Shanghai city already have the permit to provide the light freight services. Therefore, taxicabs as a new type of freight service in city distribution, also need to be focused on the reasonable fleet size.

Many scholars studied the forecasting and monitoring of taxi fleet sizes [4], and proposed that the key adjustment indicators included: The taxi ownership per 10,000 people (the threshold is 20–30 cars per 10,000 people, not less than 20 cars per 10,000 people in big cities, not less than 5 cars in small cities, and medium values in mid-sized cities); the sharing ratio of taxis in public transport (between 10–20%), mileage utilization（60% for Shenzhen, 65% for Nanjing, 70% for Dalian); the average passenger waiting time (10 and 15 minutes in off-peak and peak hours in Zhejiang—the expectation value of Ningbo city is 6.72 minutes, and the actual value is 12.49 minutes, and 3.75 minutes for Dalian expectations); the income per 100 kilometers or the average daily income of a taxi driver (related to the cost expenditure, business volume, and other factors); the traffic congestion index or the average waiting time of taxi's carrying one passenger (no clear threshold value); the average daily carrying time of a single taxi (no threshold); and other indicators.

Previous research demonstrated that there are the different index sets for adjusting the taxi fleet size in different cities; the threshold value of each indicator does not have accurate quantitative standards; and the relationships among the indicators could not be sorted by order of importance. Additionally, most of the thresholds for adjusting the taxi fleet are judged by the experience of the Department of Transportation. A scientific and rational adjustment method should be established for the decisions of taxi fleet size. Meanwhile, existing studies did not consider the novel competitive relationship with the ride-hailing taxis, such as Didi and Uber.

Therefore, this paper extracted the data of different indicators from the cruising taxicab GPS starting and ending points, trajectories, and the income data of the taxi meter. The taxi volume, average daily carrying time of the taxi, mileage utilization, driver income, average working hours per taxi per day, average passenger waiting time, sharing ratio of ride-hailing taxi, and other indicators were calculated. Then the decision tree model was applied to establish an adjustment method of the cruising taxi fleet size. The threshold values and the importance degree of the indicators were calculated using the method. Finally, the dynamic adjustment mechanism for the cruising taxis was proposed by combining the key indicators and other influential factors. The paper has two contributions: (a) Analyzed the endogenous factors determining the fleet size of the cruising taxi based on the operating datasets; (b) combined exogenous indicators based on the flied dataset into the dynamic adjustment method of taxi fleet size. The results provided a theoretical basis and decision support for dynamic adjustments of the cruising taxi fleet size.

## 2. Literature Review

Douglas first discussed the price and number regulation of the taxicab market [5]. Later authors established their models on the basis of the theory proposed by Douglas [6,7]. GPS datasets and GIS

platforms were applied to simulate the network model of taxicab services [8,9]. De analyzed the capacity utilization of the taxi market under alternative regulatory restraints and proposed the waiting time as a measure of the service quality [10]. Beesley and Glaiste observed that income could not be used independently as an indicator of changes in the number of taxis [11]. Considering the impact of taxi fleet size on passenger waiting time, Wong, K.I., Wong, S.C. and Yang proposed a network equilibrium model, which was based on a case study for the calibration and validation of the taxi model for the Hong Kong situation [12].

Yang, Ye and Tang proposed a model to determine the taxi service intensity, utilization rate, and all-day service quality level equitably, while considering the congestion externalities [13]. Schaller analyzed the on-call order market and the approaches of entry control, as well as the influence on taxi availability, including the entry qualifications and service requirements [14]. Alonso, Samaranayake, Wallar, et al. [15] presented a more general real-time and large-capacity mathematical model for ride-sharing services, which generated optimal routes based on the vehicle positions and online demand. The advantage of this model is its generality, as it can solve many multi task assignment problems.

With the appearance of riding-sharing and on-demand services, the impact of shared ride-hailing services on traditional taxicabs became a new hot topic of academic study. Vazifeh [16] provided a network-based solution to the determination of the minimum number of taxis to meet the needs of all passengers without delay. The study in New York found that the fleet size could be reduced by 30% at close to optimal service levels. Bonola, Bracciale, Loreti, et al. [17] made the experimental evaluations on the delivery performances with taxi data in Rome over six months. The results demonstrated that even with fewer vehicles running in parallel for less than a day, the coverage of the city could reach 80%.

Joaquin [18] developed a diagrammatic approach to illustrate the relationship between taxi market rules and regulations. The results indicated that the deregulated industries have a unique equilibrium corresponding to monopolistic competition. LU Jian and Wang Wei [19] proposed a taxi quantity confirming method for China by considering the taxi operating mileage, effective operating mileage, average operating speed, and average daily operating time. Mustafa [20] analyzed the impact of shared autonomous taxis on traditional taxis and posited that the conversion from traditional taxis to shared autonomous taxis could reduce the fleet size by 59%.

Previous research focused on the determinant of the taxi fleet size and answered the question of how many taxis the city needs. However, there have been no suggestions for the times to adjust the taxi fleet size. The government in cities only determines when to adjust taxi fleet size by their own experience, not by the taxi operation data. Therefore, the data-driven method is required in order to effectively adjust the taxi fleet size based on the existing city needs.

## 3. Data Collection and Analysis

The data was collected from the cruising taxicab GPS starting and ending points, the trajectory, and the income data of taxi meters in Ningbo. There were 1317 sets of valid sample data from January 2014 to August 2017, of which, 856 sets were training samples, accounting for about 65% of the decision tree structure and 35% were for the model testing. As shown in Figure 1, from 2001 till now, the fleet size of the taxis has been stable, and the growth has been slow. The number of cruising taxis has remained at 4427 since 2013, without any adjustments.

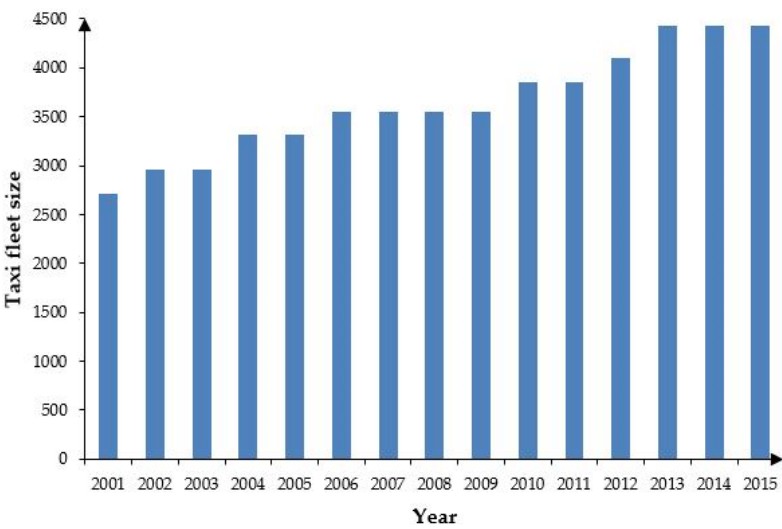

**Figure 1.** Historical numbers of taxis in Ningbo.

The distribution of key operation indicators of cruising taxis in Ningbo is shown in Table 1. The fleet size of taxis is modified by the supply–demand balance method with the actual number of taxis. The actual number of taxis was 4211 vehicles, and this changed with the time and the weather. The average daily mileage utilization rate of taxis was 0.67, and for peak and off-peak hours was 0.75 and 0.57, respectively. The average daily carrying times of a single taxi is 30.8 times. The average income per 100 km was 235 Yuan. The average daily working time of a single taxi was more than 10 hours. Due to the on-call service provided in June 2014, the number of on-call services accounted for a low proportion of total trips. More details of other indicators are shown in Table 1. Due to the collinearity issues among the indicators, the explanatory variables were determined by the effects of the indicators, which are as follows:

(1) The mileage utilization rate is denoted as the percentage of passenger mileage in the total mileage during the operating period of the taxi, which can reflect the operating status of taxi more intuitively. The average mileage utilization rate of the overall taxis can reflect whether the taxi fleet size in a city is reasonable or not. The lower the mileage utilization rate, the greater the invalid traffic flow caused by taxis in the urban road network, and the more excessive taxi fleet scale is.

(2) The average daily income of a single taxi is the most important factor for maintaining the internal stability in the taxi market. This is the most direct performance of the prosperity of the taxi industry to determine whether drivers and taxi companies should enter the industry or not. It is also a direct reflection of taxi fares.

(3) The average daily working time of a single taxi is an important indicator reflecting the working intensity of a taxi driver.

(4) The average daily carrying times of a single taxi is an intuitive indicator to describe the services of the taxi and a descriptive index of taxi passenger demand.

(5) The average waiting time to catch a single taxi is an intuitive demonstration of traffic congestion in the urban road network. It also reflects the quality of a single taxi's service.

**Table 1.** Descriptive statistics of operation indicators of cruising taxis in Ningbo.

| Indicator | Mean | Minimum | Maximum | 85% percentile |
|---|---|---|---|---|
| The actual number of taxis (vehicles) | 4211 | 1103 | 4387 | 4362 |
| The total number of trips (times) | 172,577 | 31,714 | 226,861 | 191,954 |
| The number of on-call service (times) | 1413 | 0 | 15,056 | 1649.45 |
| Total income (Yuan) | 3,848,260 | 745,433 | 4,876,810 | 4,250,160 |
| Total mileage (km) | 1,630,604 | 333,833 | 2,023,800 | 1,767,389 |
| Total operating mileage (km) | 1,097,277 | 193,486 | 1,431,900 | 1,234,745 |
| Total operating hours (minutes) | 2,549,508 | 459,440 | 4,152,456 | 2,879,151 |
| Total waiting time (seconds) | 9793 | 2091 | 15,729 | 12,679 |
| Mileage utilization ratio | 0.67 | 0.57 | 0.75 | 0.70 |
| Income per 100 kilometers (Yuan) | 235.53 | 200.64 | 262.20 | 243.88 |
| The average daily carrying times of a single taxi (times) | 30.80 | 20.26 | 45.57 | 44.66 |
| The average daily income of a single taxi (Yuan) | 910.67 | 522.03 | 1186.98 | 988.71 |
| The average wait time to catch a single taxi (seconds) | 309.49 | 227.55 | 533.02 | 354.93 |
| The average daily working time of a single taxi (minutes) | 603 | 311 | 970 | 671 |
| The forecast value of taxi fleet size (vehicles) | 4841 | 1268 | 5043 | 5014 |

As shown in Figure 2, the monthly number of cruising taxi trips changed with Didi's different marketing activities from 2014 to 2017. With the merge of the ride-hailing app Didi, the taxis were the only users of the apps and the taxi trips increased from 350,000 to 5,218,300 with the WeChat and Alibaba subsidy. However, after private car-hailing services were offered by Didi and other private ride-hailing car apps, the taxi trips decreased significantly. Until the regulations of the online ride-hailing and taxi were reformed, the ride-hailing cars were legal. After the new regulations were issued by China, the number decreased significantly and the taxi trips generally stabilized. For example, the numbers of private ride-hailing cars were 90,000 and 35,000 before and after the regulations were issued. The taxi trips fluctuated from 370,000 to 170,000.

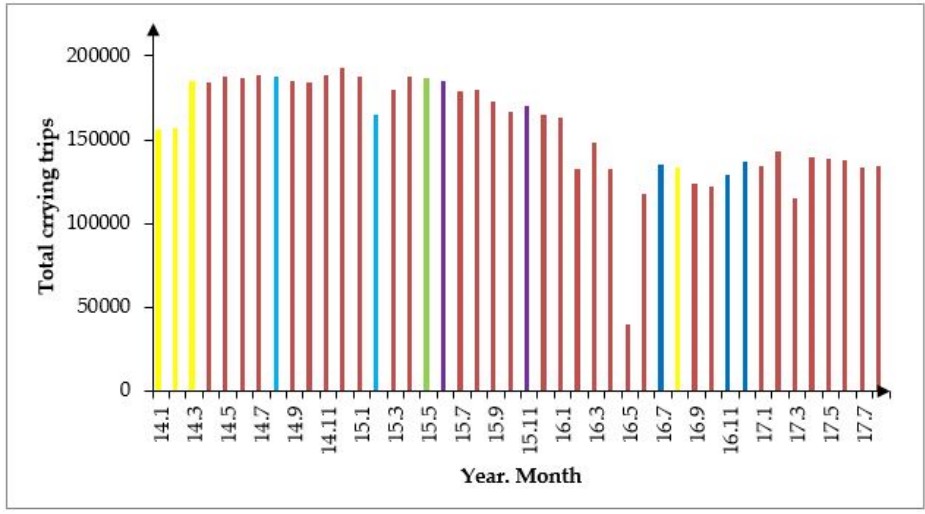

**Figure 2.** The monthly carrying trips of total taxis.

As shown in Figure 3, the average daily mileage utilization decreased from 68.70% to 58.55%, a decrease of 14.78%. The average daily carrying times of single taxi declined from 43.10 trips to 31.29 trips, a decrease of 27.40%. The revenue level per 100 kilometers decreased from 234.41 to 223.98 yuan per 100 kilometers, a decrease of 4.45%. The average daily revenue level of a single taxi dropped from

939.47 to 722.82 yuan per car, a decrease of 23.06%. With the ride-hailing taxi's rapid development, the operating efficiency of a single taxi decreased significantly. Thus, the influence of the private ride-hailing cars must be taken into consideration when the fleet size of the taxi is adjusted, and their business sharing ratio indexes are used to describe its effect.

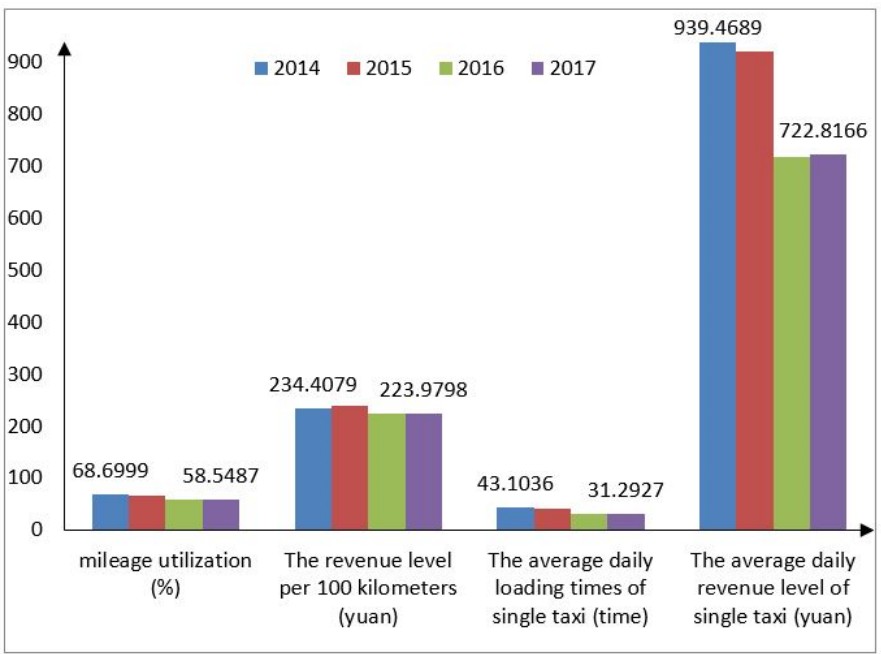

**Figure 3.** An influential analysis of online ride-hailing cars on indicators of taxis.

## 4. Methodology

### 4.1. Regression Tree Model

There are many indicators that characterize whether the fleet size for taxis is sufficient or not. It is inappropriate to adjust the taxi fleet size by using a single indicator. The relationships among the indicators should be considered to determine the fleet size of taxis. The weight and importance ranking also need to be cleared up [21]. The regression tree model used a simple and comprehensible structure for data analysis, and thus, can be used for classification, reorganization, decision making, and developing prediction models. The regression tree provides a tree structure that can be presented, easily interpreted, and defined as transparent and reproducible [22]. As an analogy, we let a key indicator like the mileage utilization be decision node m, and then $X_m$ is a subset of $X$ to node $m$, that it is all $x$ that satisfies all the decision point conditions from the root to the node m of $x \in X$, then

$$b_m(x) = \begin{cases} 1, & if\ x \in X_m : x\ \text{reach node}\ m \\ 0, & \text{otherwise} \end{cases} \quad (1)$$

The mean square error of the estimated value is the classification measure, and $g_m$ is the estimated value in the node $m$,

$$E_m = \frac{1}{N_m} \sum_t (r^t - g_m)^2 b_m(x^t) \quad (2)$$

where $N_m = |\chi_m| = \sum_t b_m(x^t)$ and $t$ is the data ID belong to set $T$.

For the node, the mean value of the output is required to reach the instance of this node:

$$g_m = \frac{\sum_t b_m(x^t) r^t}{\sum_t b_m(x^t)} \tag{3}$$

$E_m$ denotes the variance of *m*. If the error is acceptable ($E_m < \theta_r$), a tree node will be created to store $g_m$; otherwise, the data of node *m* can be further divided, making the sum of the error of the branch minimum. For each node, we found the classification threshold of the attribute of the minimizing error and numeric value, and then the above process was performed recursively.

Let $X_{mj}$ be a subset of the branching *j* for $X_m : \bigcup_{j=1}^{m} X_{mj} = X_m$ , we define

$$b_{mj}(x) = \begin{cases} 1, & \text{if } x \in X_{mj}: x \text{ reach node } m \text{ and choose branch } j \\ 0, & \text{otherwise} \end{cases} \tag{4}$$

Let $g_{mj}$ be the estimated value of the branch *j* that reaches the node *m*, then

$$g_{mj} = \frac{\sum_t b_{mj}(x^t) r^t}{\sum_t b_{mj}(x^t)} \tag{5}$$

The error after division is:

$$E'_m = \frac{1}{N_m} \sum_j \sum_t (r^t - g_{mj})^2 b_{mj}(x^t) \tag{6}$$

For arbitrary division, the reduction of the error is given by the difference between the Equation (2) and the Equation (6). The division direction is the reduction of the maximization error or equivalent to take the minimum of the Equation (6). The mean square error is a possible error function.

$$E_m = \max_j \max_t | r^t - g_{mj} | b_{mj}(x^t) \tag{7}$$

The maximum error can be used to ensure that the error of any instance is not greater than the given threshold. The acceptable error threshold is a function of complexity; the smaller the value, the larger the tree and the greater the risk of overfitting; the greater the value, the greater the likelihood of insufficient fitting and excessive smoothness.

When each decision node uses all key indicators as the input dimensions, the linear multivariable node is defined as:

$$f_m(x) : w_m^T x + w_{m0} > 0 \tag{8}$$

The instances are further divided from successive nodes on the path from the root to the leaf, while the leaf nodes define polyhedrons on the input space, fine-tune the weights $w_{mj}$, one by one, to reduce the statistical significance indicators, through subset selection to reduce the dimension and the complexity of the node.

## 4.2. Model Algorithm

According to the characteristics of the taxi fleet size, the data types of the variables are numeric, and the Chi-squared Automatic Interaction Detector of variance analysis is used as the algorithm to construct the regression tree. The model risk assessment index of the regression tree is mainly the variance and misjudgment rate of the predicted value. The standard error $\alpha_w$ of the risk trajectory is:

$$\alpha_w = \sqrt{\frac{p(1-p)}{N}} \tag{9}$$

where *p* is risk estimate, and *N* is the sample size.

The maintenance method is applied to evaluate the performance of the regression tree model and to divide the labeled raw data into two disjoint sets, which are the training set and test set. The training dataset was applied for the model calibration, and the test dataset was applied for our evaluation of the model performance.

## 5. Structure Analysis of the Decision Tree Model

In order to build the regression tree model, we used the MATLAB programming language. The suggested interval software for the maximum "tree depth" and the number of intervals were [2–10], and [1–10], respectively. The higher the depth, the more complicated the model becomes and the harder the production of the tree becomes. Low tree depth means a lower accuracy where some parameters may be omitted. Hence, the related tree depth was reduced to [3–8], and regression tree models with different controlling parameters were created by trial and error.

To evaluate the performance of the models, we calculated the root mean square error (RMSE) and used it to quantify the errors generated by each model. The values of the maximum tree depth, the number of the intervals, the minimum number of the parent node, the minimum number of the child node, and the gain ratio related to the best model are 3, 3, 1, 2, and 0.001, respectively. The confidence level of the best model is 87.36%, which indicates that the adjustment threshold value of the taxi fleet size is well summarized and explained. Figure 4 illustrates the preferable tree. The developed tree model has 20 nodes that are specified by squares and their related numbers, and the name of the variables related to the node and interval changes are shown. The correlation coefficient ($R^2$) between the calculated and predicted fleet size by the regression tree model is 0.91.

The mileage utilization has the strongest effect on the fleet size of taxis, and the F statistic was 63.73; followed by the average daily revenue of a single taxi, the average waiting time to catch a single taxi, the average operating time of a single taxi, and the revenue per 100 kilometers. The F statistics decreased in the order, indicating that the effect of the indicators on the fleet size of taxis decreased. Finally, the average daily carrying times of a single taxi was chosen as the third layer of classification indicators. The p-value of each indicator in each layer was less than 0.05 and passed the significance test. The results of the tree structure analysis revealed the relationships, interactions, and the importance order among the influential indicators. Each threshold for each indicator in the tree structure clearly provided the adjustment criteria of the taxi fleet size.

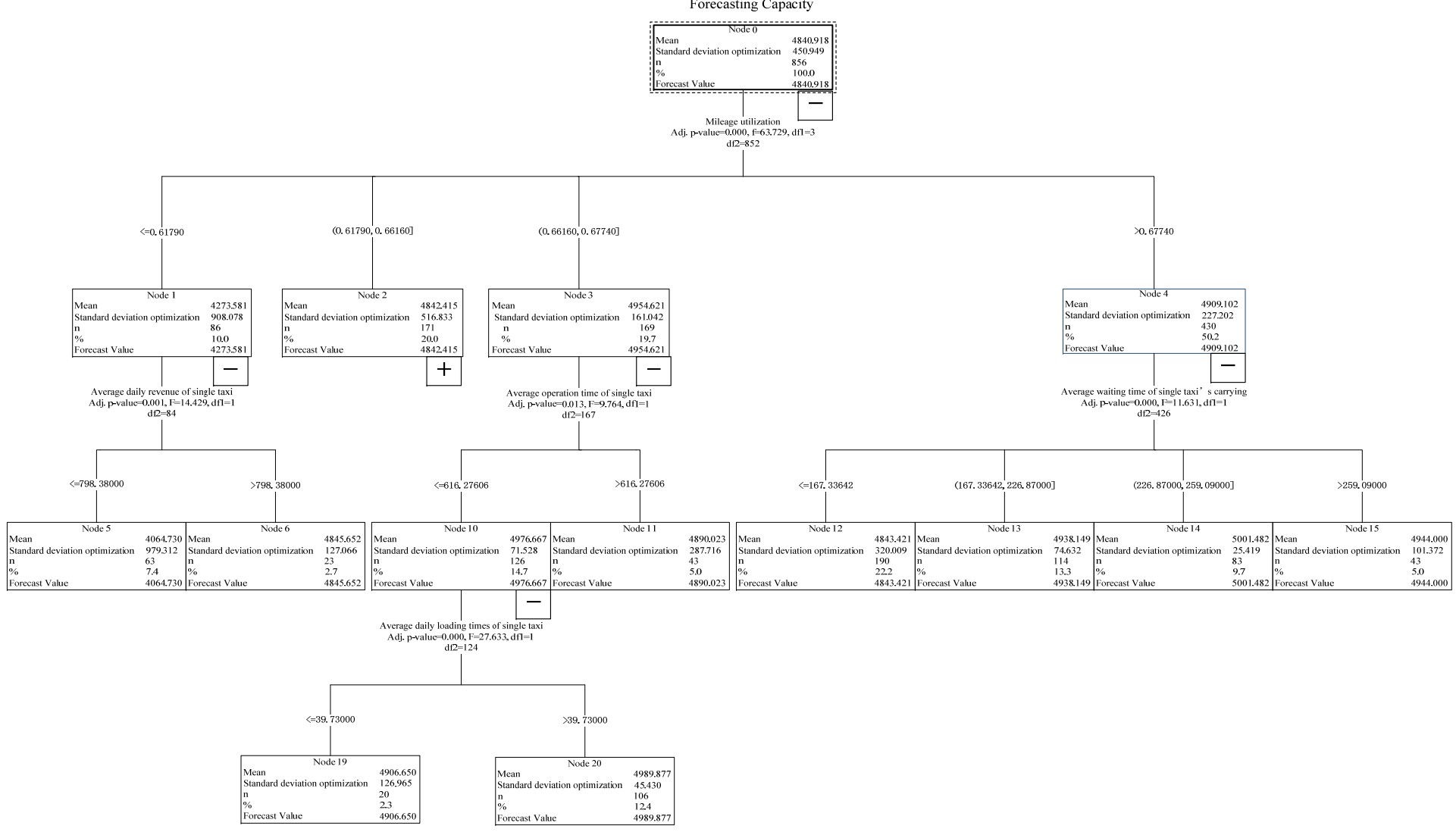

**Figure 4.** The regression tree structure for the taxi fleet size.

As the results of the regression tree model, 14 "IF-THEN" rules were generated, as shown in Table 2. Combined with the influence of taxi price on taxi demand, the typical classifications of taxi fleet size adjustment are summarized as follows:

**Table 2.** The main categories extracted from the regression tree model.

| Decision node | IF | THAN | F statistics | Classified |
|---|---|---|---|---|
| 5 | （Mileage utilization ≤ 0.6179）AND（average daily revenue of single taxi ≤ 798.38） | 4065 | 14.43 | The actual number of taxis is more than the fleet size and should be reduced by 362 vehicles |
| 6 | （Mileage utilization ≤ 0.6179）AND（average daily revenue of single taxi >798.38） | 4845 | 14.43 | The price should be reduced to increase the passenger demand |
| 7 | （0.6179 < Mileage utilization ≤ 0.6616）AND（revenue per 100 kilometer ≤ 225.01） | 4780 | 9.95 | The price should be reduced to adapt to the low-price market |
| 16 | （0.6179 < Mileage utilization ≤0.6616）AND（225.01 < revenue per 100 kilometer ≤ 235.8）AND（average daily carrying times of single taxi ≤ 37.81） | 4883 | 8.43 | The price should be reduced to attract passenger demand |
| 17 | （0.6179 < Mileage utilization ≤ 0.6616）AND（225.01 < revenue per 100 kilometer ≤ 235.8）AND（37.81 < average daily carrying times of single taxi ≤ 39.73） | 4962 | 8.43 | In the equilibrium of demand and supply, No adjustment necessary |
| 18 | （0.6179 < Mileage utilization ≤ 0.6616）AND（225.01 < revenue per 100 kilometer ≤ 235.8）AND（average daily carrying times of single taxi > 39.73） | 4994 | 8.43 | The price should be reduced to decrease the passenger demand |
| 9 | 0.6179 < Mileage utilization ≤ 0.6616）AND（revenue per 100 kilometer > 235.8） | 4437 | 9.95 | The price should be increased to adapt to the high price market |
| 19 | （0.6616 < Mileage utilization ≤ 0.6774）AND（average operating time of single taxi ≤ 616.28）AND（average daily loading times of single taxi ≤ 39.73） | 4907 | 27.63 | In the equilibrium of demand and supply, No adjustment necessary |
| 20 | （0.6616 < Mileage utilization ≤0.6774）AND（average operating time of single taxi ≤ 616.28）AND（average daily loading times of single taxi > 39.73） | 4990 | 27.63 | The price should be reduced to decrease the passenger demand |
| 11 | （0.6616 < Mileage utilization ≤ 0.6774）AND（average operating time of single taxi > 616.28） | 4890 | 9.764 | The actual number of taxis is less than the fleet size and should be increased by 463 vehicles |
| 12 | （Mileage utilization ≥ 0.6774）AND（average waiting time to catch a single taxi ≤ 167.34） | 4715 | 11.62 | In the equilibrium of demand and supply, No adjustment necessary |
| 13 | （Mileage utilization ≥ 0.6774）AND（167.34 < average waiting time to catch a single taxi ≤ 226.87） | 4807 | 11.62 | The waiting fee should be added into the price |
| 14 | （Mileage utilization ≥ 0.6774）AND（226.87 < average waiting time to catch a single taxi ≤ 259.09） | 4869 | 11.62 | Increase the waiting fee continuously and announce the adjustment warning |

| 15 | （Mileage utilization ≥ 0.6774）AND（average waiting time to catch a single taxi > 259.09） | 4812 | 11.62 | The actual number of taxis is less than the fleet size and should be increased by 385 vehicles |
| --- | --- | --- | --- | --- |

(1) For node 5, the mileage utilization is low, which indicates that the taxi demand is less than the supply and the actual number of taxis is more than the number that satisfies the demand. On the other hand, the average daily revenue of a single taxi is very low, which indicates that the driver's income is low and they will not be willing to provide taxi service any more. Therefore, the actual number of taxis is far in excess of the taxi fleet size, and needs to be reduced in capacity (4427 – 4065 = 362 vehicles) to achieve a balance between the taxi supply and demand.

(2) For node 6, the mileage utilization is low, but the average daily revenue of a single taxi is in the high condition, which indicates that the driver's income is acceptable. The fleet size of taxis does not need to be adjusted, but the price of taxis should be reduced to attract passenger demand and improve mileage utilization.

(3) For node 7, the mileage utilization is within the range of (0.6179, 0.6616), but the revenue per 100 kilometers is less than 225.01. The income of the taxis is slightly less than the actual average price (2.35 yuan/km), which shows that the driver's income is unacceptable. However, the fleet size of the prediction value in this child node is slightly smaller than the parent node 2, so there is no need to adjust the number of taxis. However, the price should be reduced to aim to improve the driver's revenue situation and adapt to the low-price market.

(4) For node 19, the mileage utilization is within the range of (0.6616, 0.6774), the threshold value is high, which indicates that the demand is slightly higher than the supply, but the average operating time of a single taxi is less than or equal to 616.28, which indicates that the driver's working strength does not exceed the limit of tolerance. Moreover, the average daily carrying times of a single taxi is less than or equal to 39.73, which does not exceed the limit value. Therefore, the difference between the demand and supply is acceptable, and there is no need to adjust the fleet size of taxis to achieve the balance between the supply and demand.

(5) For node 15, the average waiting time to catch a single taxi exceeds the passenger's tolerance, which shows that the traffic condition deteriorated drastically and the taxi pays extra costs for completing the service under the conditions of traffic congestion. At the same time, the mileage utilization is so high that passenger demand could not be satisfied with the actual supply of taxis. Therefore, the fleet size of taxis should be increased (4812 – 4427 = 385 vehicles) to reduce the mileage utilization, satisfy passenger demand, and compensate for the extra cost caused by the traffic congestion.

In addition, the cumulated distributions of the predicted values from the training dataset and testing dataset are shown in Figure 5. The variance and misjudgment rate of the estimated values were used to evaluate the risk of the regression tree model. The standard error of the training and testing dataset was 6.13%, and the misjudgment rate was 0.07. The overall accuracy of the model was better. The classification results can be used to guide the adjustment of the city's capacity, as in theory the demand during peak and off-peak hours should be able to be improved and met with greater efficiency than the current market performance.

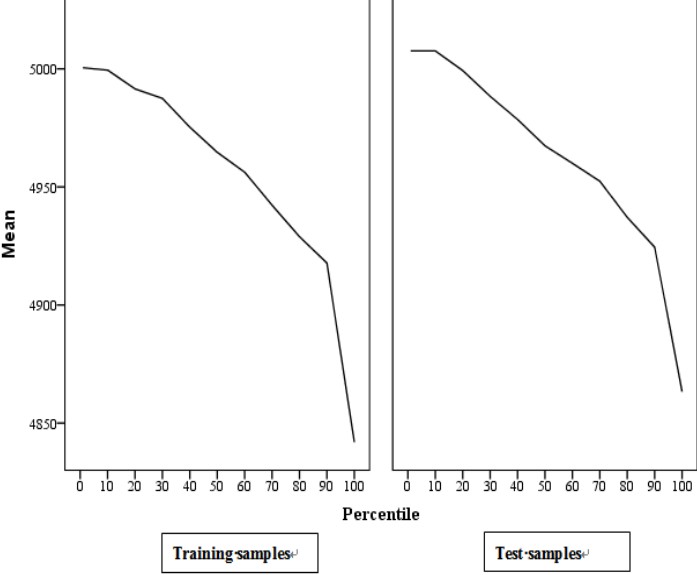

**Figure 5.** Cumulated distributions of the training and testing samples.

## 6. Dynamic Adjustment Mechanism of Taxis Fleet Size

The objective of the taxi fleet size adjustment was to achieve the balance between supply and demand and reduce the taxi-hailing difficulty. The endogenous indicators were summarized in the regression tree model. However, the exogenous indicators were not considered in the model, and also had a significant influence on the adjustment of the taxi fleet size. The difference between the exogenous and endogenous indicators is the dataset. The exogenous indicators could not be collected in the operating datasets. As they reflect different relationships between the taxi fleet size and other influential factors, it is necessary to combine them into the dynamic adjustment mechanism.

Taking the business ratio between cruising taxis and ride-hailing taxis, if the number of ride-hailing taxis increases dramatically, more and more passengers would switch from cruising taxis to the ride-hailing services. The cruising taxi market would be destroyed. Therefore, the proposed mechanism put the exogenous indicators in front of the endogenous indicators. Meanwhile, the importance of the exogenous indicators was also sorted by the Delphi method. Then, the proposed mechanism combined the exogenous and endogenous indicators in the unified decision procedure. According to the existing literature regarding the taxi regulation methods, the exogenous indicators were as following:

(1) The taxi ownerships per 10,000 persons directly reflected the relationship between the scale of the taxi capacity and the urban population. This is the earliest index for evaluating the status of the supply and demand equilibrium of the taxis to adjust the number of taxis in an urban area. For example, the Ministry of Transport of China regulated that the taxi ownerships per 10,000 persons should not be less than 25 taxis per 10,000 persons for a city of more than 100 million people. Although this indicator has no obvious significance, due to its lack of demonstrating taxi market performance, it is the most attentive value in the minds of the public and the media. The data of the taxi ownerships per 10,000 persons in many cities in China were collected. Compared with the other cities with a similar level of population, economic, and social development, the threshold of the taxi ownerships per 10,000 persons is suggested at 23 vehicles per 10,000 people recently in Ningbo. Please see Figure 6.

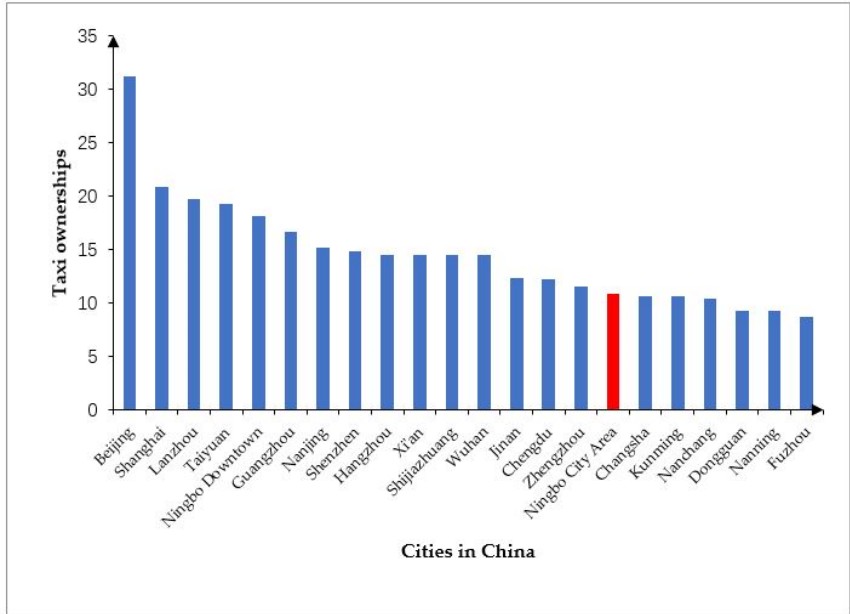

**Figure 6.** Taxi ownerships per 10,0000 persons of different cities in China.

(2) The sharing ratio of taxis in public transport reflects the relationship between taxi traffic and transit transport. The higher the sharing ratio of taxis is, the lower the level of service of transit traffic will be. According to the transit planning requirement, the optimum value of the taxi sharing ratio in public transit is changing from 8% to 20% for different cities. Based on the analysis of the traffic mode split rate in the latest Origin-Destination survey in Ningbo, the threshold of this indicator is 18%.

(3) The business ratio between the taxi and ride-hailing car describes the impact of the online ride-hailing car on the dropping effect of the cruising taxi trip. As shown in Figure 7, the comparison of business volume between the ride-hailing and cruising taxis was described. The business volume of the cruising taxi and online ride-hailing taxis reached a steady state until the regulations of the online ride-hailing taxi were proposed in June 2017. Therefore, the steady ratio of 10:6.54 is used as the threshold value of the sharing ratio between the cruising and ride-hailing taxis.

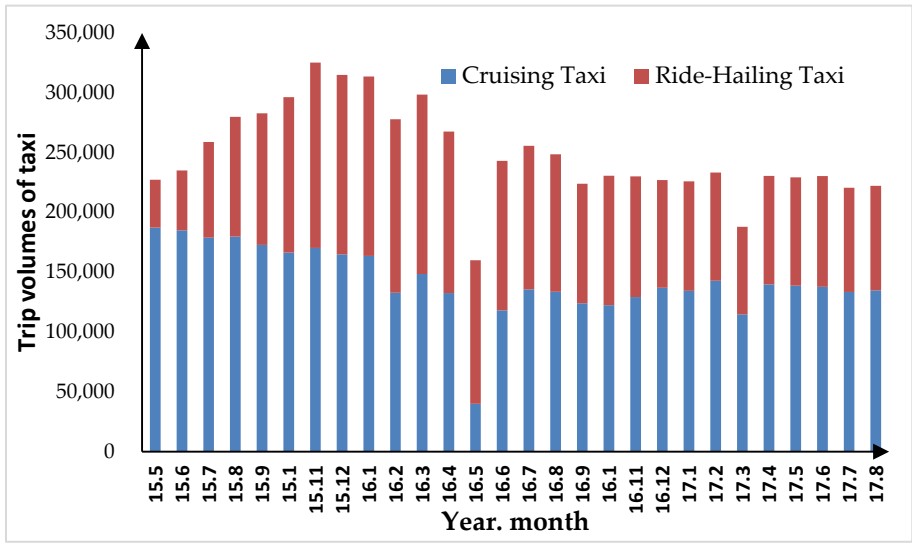

**Figure 7.** Comparison analysis on the trip volumes of ride-hailing and cruising taxis.

(4) The average waiting time of passengers reflects the satisfaction of the taxi service and the status of supply and demand. The higher passenger waiting time, the lower the satisfaction of the taxi service is. This also means that the supply of taxis is less than passenger demand. The average waiting time of passengers was investigated in different regions (Colleges and Universities; Large shopping districts; Large parks; Tourist attractions; Transport hub; Subway transfer station; Large hospitals; Randomly selected locations in addition to the above areas).

As shown in Figure 8, the average waiting times of the passenger who calls by the side of the road and on-call through the network were 13.53 and 11.72 minutes at the peak hour, and 8.23 and 5.87 minutes at off-peak hours. The threshold values of the average waiting time of the passenger at the peak and off-peak were 12.82 and 9.61 minutes, respectively. Considering the characteristics between the supply and demand at peak and off-peak hours, the threshold value of the average passengers waiting time should be calculated at 11.38 minutes using the demand elasticity (1.1) at the peak and the supply utilization rate (0.9) at the off-peak.

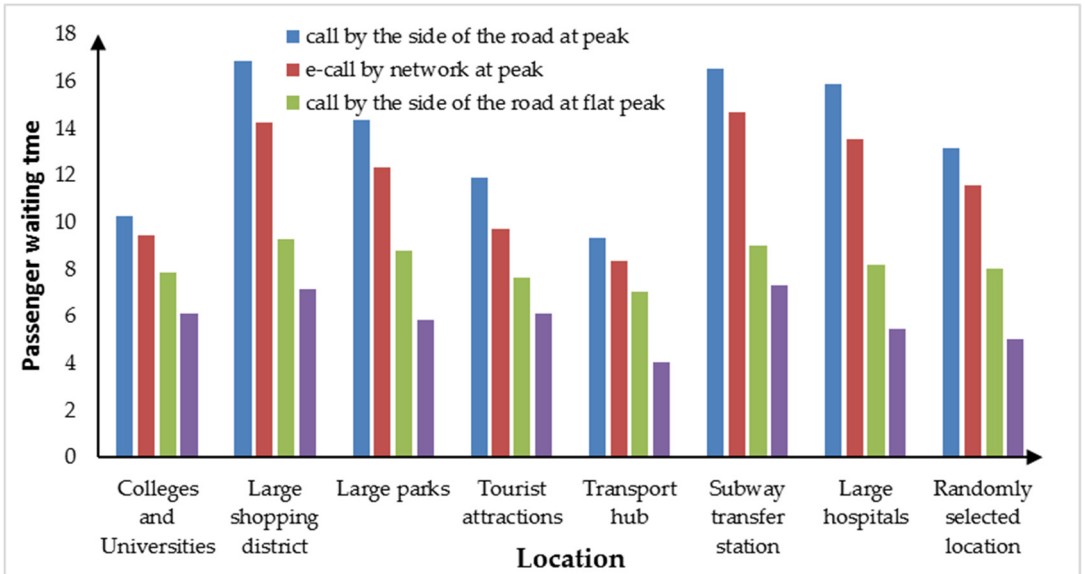

**Figure 8.** Distributions of the average waiting times of passengers for cruising and online ride-hailing taxis at different places and times.

Additionally, the importance of the exogenous and endogenous indicators was sorted by the Delphi method. As shown in Figure 9, the dynamic adjustment mechanism of the taxi fleet size was designed in combination with various indicators. The rank and threshold value of each indicator was also given. The adjustment mechanism could be introduced to identify the disequilibrium of supply and demand and determine the taxi fleet size for the city.

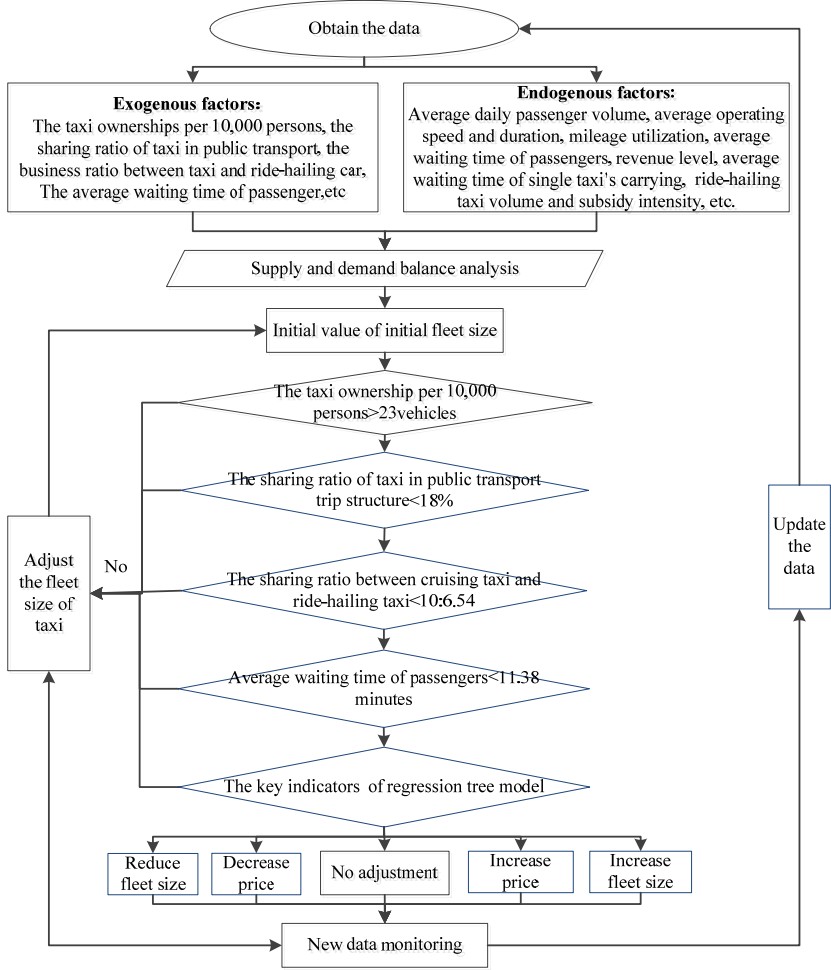

**Figure 9.** The dynamic adjustment mechanism of the scale of cruising taxi capacity.

Taking Ningbo as an example, the survey data and operating data of taxis were collected. The dynamic adjustment method was applied as follows: The current fleet size of taxis was 4427 vehicles. The taxi ownership per 10,000 persons was 24, higher than the 23 taxis per 10,000 persons. Based on the latest public transport survey, the sharing ratio of taxis in public transport was 17.6%, lower than 18%. The average business volumes of the cruising and online ride-hailing taxis were 133,578 and 88,161. The business sharing ratio between the taxis and ride-hailing cars was 10:6.60. From the survey, the average passenger waiting time was 10.29, lower than 11.38 minutes.

The latest operating taxi data was applied to calculate the indicators of the regression tree model. The result demonstrated that the current taxi fleet size is not sufficient and should be increased by 338 taxis. Compared to the other methods discussed in the Introduction section, and shown in Table 3, the taxi fleet size should increase on average around 1251 cars. Clearly, this is not practical with the increase of the ride-hailing taxi service. More and more drivers switched from cruising taxis to ride-hailing taxis in the taxi market. Only 300 new energy taxis were put into the market by the government, due to low cost. Therefore, the proposed mechanism was valid and verified by the real market.

**Table 3.** Comparison with other methods.

| Single Indicator | Threshold Value for Indicator | Calculation Methods | Fleet Size of Taxicab | Adjustment Result |
|---|---|---|---|---|
| The taxi ownership per 10,000 persons | 21 taxis per 10,000 persons | total population size of 2.3012 million × 25 taxis per persons | 5753 | 1326 |
| Mileage utilization | 65% | $$N = \frac{L}{T \cdot (1-K) \cdot V \cdot 0.9}$$ where $N$ is fleet size; $L$ is total valid taxi miles, 1.5380 million km; $T$ is the work time for a taxi, 13 hours; $V$ is the average operating speed of taxi, 28 km/h. | 5721 | 1294 |
| The average passenger waiting time | 6.72 minutes | $$N = \frac{L \cdot f(w_t)}{T \cdot (1-K) \cdot V \cdot 0.9} \text{ and}$$ $$f(w_t) = \frac{w_t^e}{w_t^c}$$ where $w_t^e$ is the expected waiting time, 6.72 minutes; $w_t^c$ is the current waiting time, 6.75 minutes. | 5707 | 1280 |
| The average daily income of taxi driver | No standards. The average daily income of Nignbo city, 693 Yuan per day, was recommended as the threshold value. | $$N = \frac{L \cdot f(I_d)}{T \cdot (1-K) \cdot V \cdot 0.9} \text{ and}$$ $$f(I_d) = \frac{I_d^c}{I_d^e}$$ where $I_d^c$ is the current average daily income, 718 Yuan per day and $I_d^e$ is the expected daily income, 693 Yuan | 5534 | 1107 |

## 7. Conclusions

Due to the lack of adjustment index system of taxi fleet size, the taxi operating datasets were collected from the cruising taxicab GPS starting and ending points, trajectories, and the income data of taxi meters in Ningbo. A regression tree model was applied to establish the adjustment method of taxi fleet size by considering the endogenous indicators, which contained the mileage utilization, average daily loading times of a single taxi, income per 100 km, the average operating time of a single taxi, the average waiting time to catch a single taxi, and other operating indicators of taxis. The endogenous indicators concerning the ownerships per 10,000 persons, the sharing ratio of taxis in public transport, the business sharing ratio between taxis and online ride-hailing cars, and the average waiting times of passengers were also brought to classify the taxi fleet size. The importance of the exogenous and endogenous indicators was sorted by the Delphi method. The dynamic adjustment mechanism of the taxi fleet size was designed by combining the exogenous and endogenous indicators. The rank and threshold value of each indicator was also given. The results indicated that:

(1) In the three-layer structure of the regression tree model, the mileage utilization has the strongest effect on the fleet size of taxis, and the F statistic was 63.73; followed by the average daily revenue of a single taxi, the average waiting time to catch a single taxi, the average operating time of a single taxi, and the revenue per 100 kilometers. The *p*-values of the indicators in every layer were less than 0.05 and passed the significance tests. The standard error of the training and testing dataset was 6.13%, and the misjudgment rate was 0.07. The overall accuracy of the model was better.

(2) When the mileage utilization ratio is less than 0.6179 and the average daily revenue of a single taxi is less than 798.38 Yuan, the actual number of taxis is far in excess of the suggested taxi fleet size, and the capacity needs to be reduced (4427 − 4065 = 362 vehicles) so as to achieve the balance between taxi supply and demand.

(3) When the mileage utilization is between 0.6616 and 0.6774 and the average operating time of a single taxi is more than 616.28 minutes, the fleet size of taxis is insufficient and needs to be increased by 385 vehicles.

(4) The threshold value of taxi ownership per 10,000 persons, the sharing ratio of taxis in public transport, the sharing ratio between taxis and online ride-hailing cars, and the average waiting time of passengers are 23 vehicles per 10,000 persons, 18%, 10:6.54, and 11.38 minutes, respectively.

A reasonable dynamic adjustment of taxi fleet size could effectively alleviate the conflicts between taxis and online ride-hailing cars and promote the green and sustainable development of the taxi industry and urban transportation. For the future, it is necessary to model the whole of taxi services on the passenger network [23], and use innovative equipment to alleviate the contradiction between cruising and ride-hailing taxis [24]; and thus, obtain more reasonable decision support for adjusting taxi fleet sizes [25]. The appearance of shared autonomous vehicles will again challenge the taxi market and impact taxi fleet sizes [26].

**Author Contributions:** The authors confirm contribution to the paper as follows: study conception and design: M.L., X.Y. (Xiaofei Ye); data collection: X.Y. (Xingchen Yan); analysis and interpretation of results: M.L., X.Y. (Xiaofei Ye), J.C.; draft manuscript preparation: X.Y. (Xiaofei Ye), Z.Y. All authors reviewed the results and approved the final version of the manuscript.

**Funding:** Natural Science Foundation of Zhejiang Province, China Grant number (No. LY20E080011), Natural Science Foundation of China (No.71971059, 71701108 and 71861006), National Key Research and Development Program of China—Traffic Modeling, Surveillance and Control with Connected and Automated Vehicles, China Grant number (No.2017YFE9134700) ,Natural Science Foundation of Ningbo, Zhejiang Province, China Grant number (No. 2017A610139) Basic Research Program of Science Grant number (No.BK20180775) and Technology Commission Foundation of Jiangsu Province, China Grant number (No.BK20170932]).

**Acknowledgments:** The authors thank the relevant institutions for funding this project. The authors thank drivers for providing this work data.

**Conflicts of Interest:** The authors declare no conflict of interest.

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
