# Peer review of "A Dynamic Adjustment Model of Cruising Taxicab Fleet Size Combined the Operating and Flied Survey Data"

_sustainability, doi:10.3390/su12072776_

Round 1

Reviewer 1 Report

The paper studies the taxi system in Ningbo City (China). The aim is to dynamically adjust the taxi fleet size. The data from the company that operates the taxis is analyzed. The paper is well written, however I do not see the contributions. Some already existing method are explained and applied. For me it is a good engineering study, but not sufficient to be published in a research journal.

Reviewer 2 Report

  1. This paper has two contributions: a) analyzing factors affecting fleet size of taxicab based on regression tree model, b) proposing dynamic adjustment. However, the title only include the latter. I think the title should include both.

  1. Is there any previous research about dynamic assignment of the fleet size? If so, it should be further discussed in literature review, and the title should be “A Dynamic Adjustment Model ~”.

  1. The structure of the paper is not well organized. Most of contents of analysis is about regression tree model. The discussions of the dynamic adjustment mechanism are very brief. It should be more detail.

  1. How the proposed mechanism can be valid? Authors proposed the mechanism in figure 9. However, I cannot find any result of them.

  1. In line 283-285 page 13, authors mentioned that “The endogenous indicators were summarized in the regression tree model. But exogenous indicators were not considered in model and also have significant influence on the adjustment of taxi fleet size.” What is the empirical effect of exogenous indicators? How it is different between when we only consider endogenous indicators (regression tree model) and when we consider both (with the proposed mechanism)?

Reviewer 3 Report

The article is very promising, but requires the following comments.

Is it a research question?: ”How to regulate the reasonable fleet size of the cruising taxi to satisfy the traffic demand of the residents without sacrificing the income of taxi driver is an important premise of taxi police formulation.” It should be commented as such.

It would be interesting for Authors that running Uber in countries such as Finland is not allowed. 

References should be given in square brackets.

”CH J D C and M J B 16”? - it seems incorrect.

The graphs in fig. 2, 3 are illegible and missing units occur. Moreover, it is not possible to acquaint with selected values given in fig. 3.

Several parameters given in equations are not defined e.g. rt, xt, gm, bm among others. Sum by t suggest it is only one element. And t should be given as a set T here.

Spelling and grammar errors occur in the paper, therefore it should be checked very carefully before resubmission.

Pages numbering is incorrect as well.

Section 5. needs deeper description at the very beginning. It should correspond to the previous sections.

Regression Tree Structure of Taxi Fleet Size is very important part of the paper therefore it needs more detailed description.

The paper needs future research description. Prediction models are mentioned in the paper, therefore Authors might take in into consideration the following research: Czwajda L., Kosacka-Olejnik M., Kudelska I., Kostrzewski M., Sethanan K., Pitakaso R., 2019, Application of prediction markets phenomenon as decision support instrument in vehicle recycling sector, LogForum, Vol. 15, Issue 2, pp. 265-278. DOI: 10.17270/J.LOG.2019.329

The issue is so complex that it would be worth considering multiset topics for future research: Kostrzewski M, Kostrzewski A., 2019, Analysis of Operations upon Entry into Intermodal Freight Terminals. Applied Sciences. 2019; 9(12):2558, pp 1-15. DOI: 10.3390/app9122558

Moreover, for future research, taxis nowadays needs using innovative equipment so it can be taken into consideration. The aspects of innovative way of treating customers and applied equipment e.g. based on https://doi.org/10.1016/j.promfg.2020.01.080

Round 2

Reviewer 2 Report

Authors well discussed my previous concerns.

Author Response

Thanks again for your help.

Reviewer 3 Report

The current version of the paper is much better in several aspects. Very well done! However, Authors are asked to include additional remarks in order to make the contribution even better.

Authors are asked to check the choice of fonts and their sizes. Sometimes it does not correspond with guidelines, e.g. in second section, the last paragraph of the paper.

If Authors planned to present the trend on Fig. 2. they might also formulate its equation and compute determination coefficient.

Sum by t suggest it is only one element. And t should be given as a set T here: t ∈ T (similar with j under max function)

It is suggested to write ”passenger waiting time” instead of ”passenger wait time”.

In selected parts of Fig. 9 plural should be included ("10,000 person").

Authors are also asked to check the guidelines in case of references No. 4, 19.-25 (please check, names, surnames and big letters).
